# Comparative Molecular and Immunoregulatory Analysis of Extracellular Vesicles from *Candida albicans* and *Candida auris*

Daniel Zamith-Miranda,[a,b] Heino M. Heyman,[c] Sneha P. Couvillion,[c] Radames J. B. Cordero,[d] Marcio L. Rodrigues,[e,f] Leonardo Nimrichter,[f] Arturo Casadevall,[d] Rafaela F. Amatuzzi,[e] Lysangela R. Alves,[e] Ernesto S. Nakayasu,[c] Joshua D. Nosanchuk[a,b]

aDepartment of Microbiology and Immunology, Albert Einstein College of Medicine, Bronx, New York, USA

bDivision of Infectious Diseases, Department of Medicine, Albert Einstein College of Medicine, Bronx, New York, USA

cBiological Sciences Division, Pacific Northwest National Laboratory, Richland, Washington, USA

dHarry Feinstone Molecular Microbiology and Immunology Department, Johns Hopkins Bloomberg School of Public Health, Baltimore, Maryland, USA

eCarlos Chagas Institute, Fundação Oswaldo Cruz (Fiocruz), Curitiba, Brazil

fMicrobiology Institute Paulo de Góes (IMPG), Federal University of Rio de Janeiro, Rio de Janeiro, Brazil

Ernesto S. Nakayasu and Joshua D. Nosanchuk share senior authorship.

**ABSTRACT** *Candida auris* is a recently described multidrug-resistant pathogenic fungus that is increasingly responsible for health care-associated outbreaks across the world. Bloodstream infections of this fungus cause death in up to 70% of cases. Aggravating this scenario, the disease-promoting mechanisms of *C. auris* are poorly understood. Fungi release extracellular vesicles (EVs) that carry a broad range of molecules, including proteins, lipids, carbohydrates, pigments, and RNA, many of which are virulence factors. Here, we carried out a comparative molecular characterization of *C. auris* and *Candida albicans* EVs and evaluated their capacity to modulate effector mechanisms of host immune defense. Using proteomics, lipidomics, and transcriptomics, we found that *C. auris* released EVs with payloads that were significantly different from those of EVs released by *C. albicans*. EVs released by *C. auris* potentiated the adhesion of this yeast to an epithelial cell monolayer, while EVs from *C. albicans* had no effect. *C. albicans* EVs primed macrophages for enhanced intracellular yeast killing, whereas *C. auris* EVs promoted survival of the fungal cells. Moreover, EVs from both *C. auris* and *C. albicans* induced the activation of bone marrow-derived dendritic cells. Together, our findings show distinct profiles and properties of EVs released by *C. auris* and by *C. albicans* and highlight the potential contribution of *C. auris* EVs to the pathogenesis of this emerging pathogen.

**IMPORTANCE** *Candida auris* is a recently described multidrug-resistant pathogenic fungus that is responsible for outbreaks across the globe, particularly in the context of nosocomial infections. Its virulence factors and pathogenesis are poorly understood. Here, we tested the hypothesis that extracellular vesicles (EVs) released by *C. auris* are a disease-promoting factor. We describe the production of EVs by *C. auris* and compare their biological activities against those of the better-characterized EVs from *C. albicans*. *C. auris* EVs have immunoregulatory properties, of which some are opposite those of *C. albicans* EVs. We also explored the cargo and structural components of those vesicles and found that they are remarkably distinct compared to EVs from *C. auris*'s phylogenetic relative *Candida albicans*.

**KEYWORDS** *Candida auris*, *Candida albicans*, extracellular vesicles, multi-omics, fungal pathogenesis, candidiasis

Address correspondence to Ernesto S. Nakayasu, ernesto.nakayasu@pnnl.gov, or Joshua D. Nosanchuk, josh.nosanchuk@einsteinmed.org.

 *C*andida auris is a recently described pathogenic fungus that has emerged as a serious cause of health care-associated infections across the world (1). Therefore, it is considered a global threat by the U.S. Centers for Disease Control and Prevention (2). The biological challenges of combatting *C. auris* include the capacity of the fungus to form resilient biofilms and to resist multiple antifungal drugs (3). *C. auris* kills 30 to 70% of infected individuals (4). Although we have deep knowledge regarding the disease-promoting mechanisms deployed by other *Candida* species, relatively little is known about *C. auris*. We recently compared the molecular profiles of two *C. auris* isolates versus that of *Candida albicans* by integrating proteins, lipids, and metabolites of these yeast cells, and demonstrated that *C. auris* has an elevated expression of pathways related to drug resistance and virulence, such as sterol metabolism and drug resistance-related transporters (5).

Disease development is a combination of fungal virulence factors and the affected host's ability to efficiently control the fungal growth, and extracellular vesicles (EVs) play a role in both of these factors. EVs are lipid-bilayered structures released by a broad variety of unicellular or multicellular organisms (6). Fungal EVs from *Cryptococcus neoformans* were first described in 2007 (7), and they have since been shown to be an important mechanism for molecular export in a variety of fungal species. EVs produced by fungi carry many biologically active molecules, including virulence factors and regulators, indicating that they could activate the innate immune system and influence disease development (8–16). *In vitro*, fungal EVs impact phagocyte activity, promoting an increase in cytokine levels, modulating phagocytosis, and regulating macrophage polarization (8–10, 14, 16, 17). Together, these data strongly suggest that fungal EVs activate the immune response. Indeed, *Galleria mellonella* larvae are protected by pretreatment with EVs from *C. albicans*, *C. neoformans*, and *Aspergillus flavus* (9, 14, 18). Recently, we demonstrated that immunization of mice with EVs from *C. albicans* confers full protection against systemic candidiasis (11).

However, the outcome of fungal EV and host response depends on the model investigated. For instance, yeast EVs released from cocultures of dendritic cells (DCs) and *Malassezia sympodialis* induce the production of tumor necrosis factor alpha (TNF-$\alpha$) and higher levels of interleukin 4 (IL-4) by peripheral blood mononuclear cells (PBMCs) from patients with atopic eczema compared to control PBMCs, displaying an allergic reaction (19, 20). *C. neoformans* and *Sporothrix brasiliensis* EVs are associated with virulence and disease progress in murine models, respectively (21, 22). We hypothesize that the multiple activities attributed to fungal EVs could be dependent on their composition, which at least partially differs according to the species investigated (9, 20, 23–28). Thus, a more complete analysis of EV composition could open new views for understanding fungal diseases.

Here, we performed a detailed characterization of EVs released by two distinct strains of *C. auris* (MMC1 and MMC2, which are highly resistant and susceptible to fluconazole, respectively) (5) and *C. albicans*. Differences in size and sterol/protein ratios were observed. Using integrated multi-omics (proteomics, lipidomics, and transcriptomics) analysis, we compared EVs and whole cells of *C. auris* and *C. albicans* and demonstrated significant compositional differences that could impact pathogenesis. Developing functional assays, we demonstrated that *C. auris* EVs influence adhesion to epithelial cells, intracellular killing by macrophages, and activation of dendritic cells. Together, our results show that *C. auris* produces EVs with a composition distinct from those of *C. albicans*, and *C. auris* EVs modulate host cell defense mechanisms.

## RESULTS

**Morphological characterization of *C. auris* extracellular vesicles.** EVs were isolated from the supernatant of *C. albicans* and *C. auris* cultures and then analyzed by transmission electron microscopy (TEM). As reported previously, EVs from *C. albicans* are round and bilayered particles (Fig. 1A) (9). Similar results were observed for EVs from both *C. auris* isolates (Fig. 1B and C), consistent with the reported morphology of

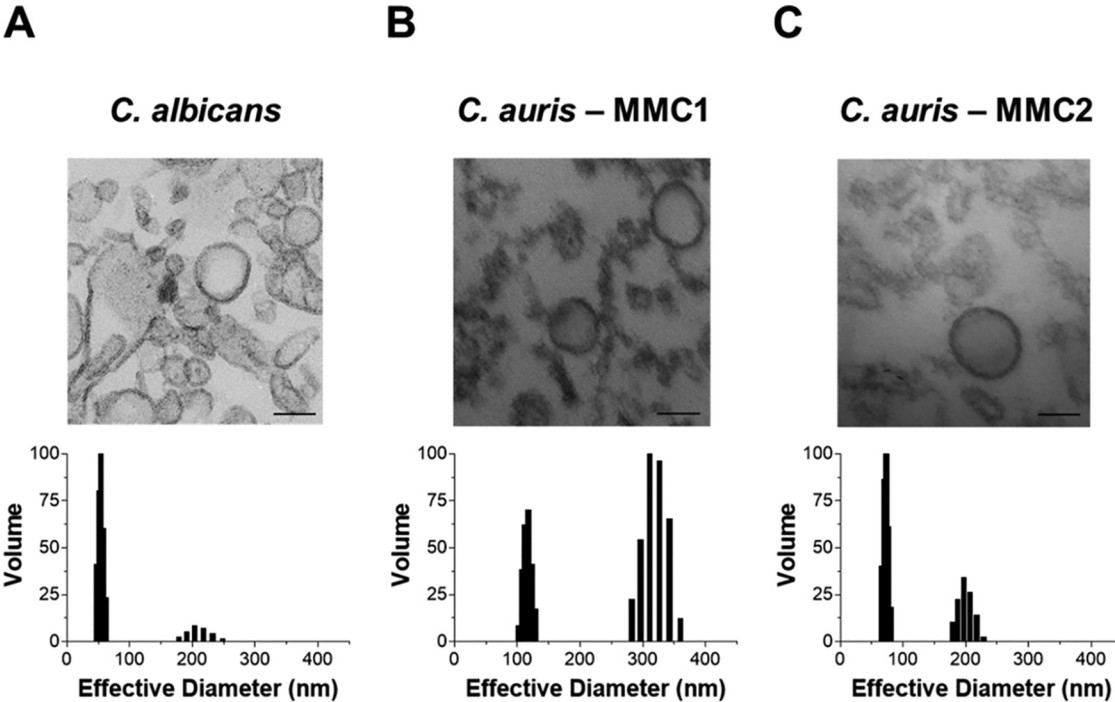

**FIG 1** *Candida auris* releases extracellular vesicles (EVs). Transmission electron micrographs and dynamic light scattering measurements of EVs from *Candida albicans* (A) and *C. auris* isolates MMC1 (B) and MMC2 (C). Two independent EV isolations were analyzed by both methods with similar results. The figure shows representative results of each analysis. Bars, 100 nm.

other fungal EVs (7, 9, 12, 16, 19, 22, 24, 29). EVs were also analyzed by dynamic light scattering (DLS) to evaluate their global size. The sizes of EVs isolated from *C. albicans* and *C. auris* MMC2 were very similar, ranging from 50 and 70 nm and, in the second population, between 170 and 250 nm (Fig. 1A and C). *C. auris* MMC1 produced EVs of larger hydrodynamic size, ranging from 100 to 150 nm and, in the second population, between 280 and 370 nm (Fig. 1B).

**Protein and ergosterol content of *C. auris* extracellular vesicles.** The amounts of protein (Fig. 2A) and ergosterol (Fig. 2B) were determined and normalized by the number of yeast cells in culture at the EV harvest time. Both *C. auris* strains secreted similar amounts of protein in EVs, but the amounts were 3 to 4 times lower than those secreted by *C. albicans*. Likewise, the amounts of EV ergosterol was 3 to 6 times lower

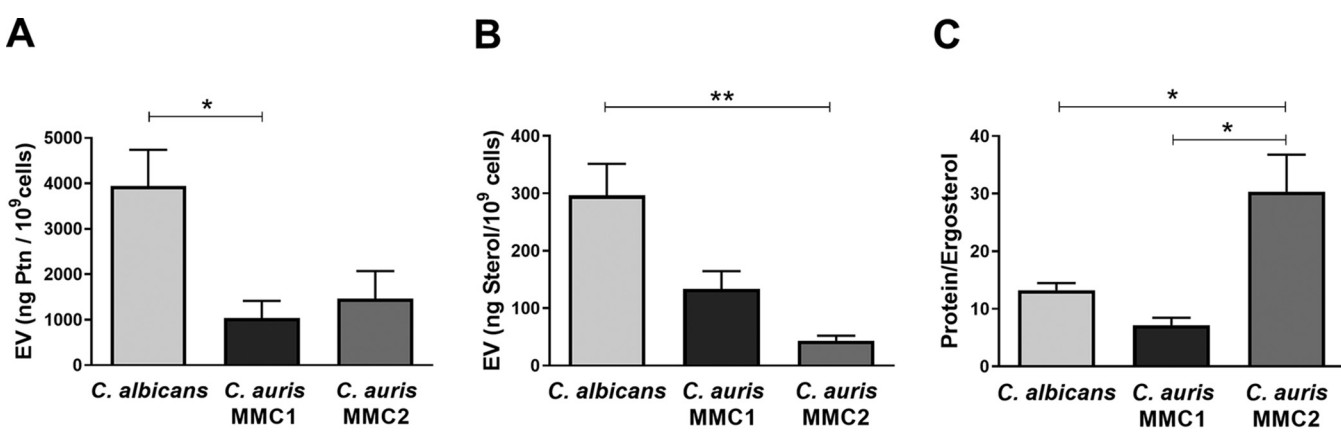

**FIG 2** Protein and sterol content in *C. auris* and *C. albicans* extracellular vesicles (EVs). Protein (A) and sterol (B) concentrations were measured in EVs suspensions from *C. auris* and *C. albicans* and normalized by the number of cells present in the fungal cultures at the harvest time. (C) Protein to ergosterol concentration ratios. All graphs represent means and standard error of the mean, relative to 4 independent EV isolations. *, $P \leq 0.05$; **, $P \leq 0.01$; one-way analysis of variance (ANOVA) followed by Tukey's multiple-comparison test.

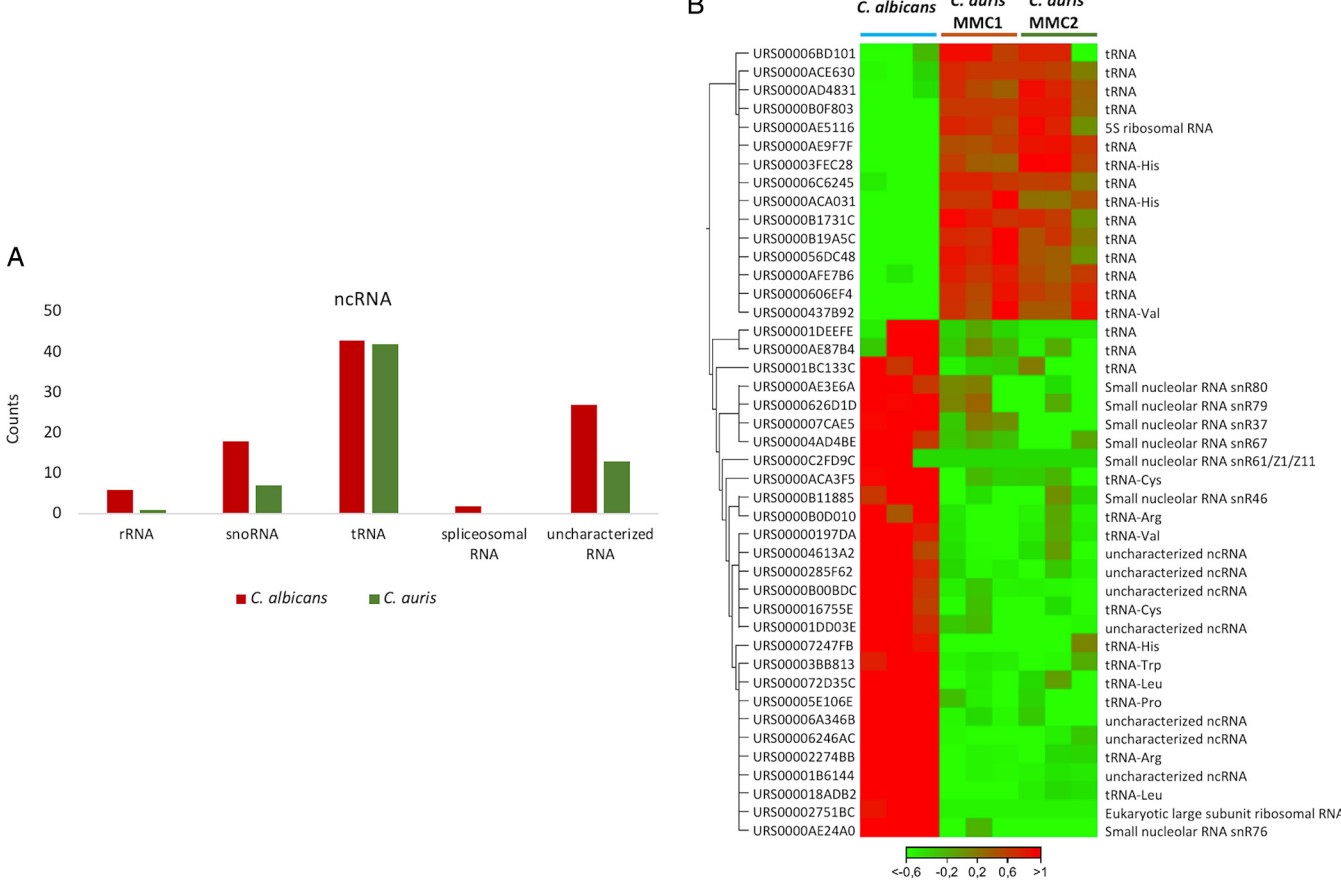

**FIG 3** EV from *C. auris* carry RNA. (A) Comparison between noncoding RNAs (ncRNAs) enriched in the *C. auris* EVs compared to *C. albicans* EVs. (B) Heatmap representing differentially expressed ncRNAs between the EVs of *C. auris* and *C. albicans* (false-discovery rate [FDR], <1%; fold change [FC], 10). The expression levels are visualized using a gradient color scheme, as follows: blue, high expression level; red, low expression level. Values represent the logFC.

in *C. auris* strains than those in in *C. albicans*. *C. auris* MMC2 strain EVs had the smallest amount of ergosterol, 3 times lower compared than that in MMC1 EVs (Fig. 2B). As ergosterol is a ubiquitous molecule present in EV membranes, we normalized the protein content by the amount of ergosterol in each strain as a way to measure possible differential protein loads among isolates. MMC2 had a higher protein/ergosterol ratio than either MMC1 or *C. albicans*.

**RNA content of *C. auris* extracellular vesicles.** We performed next-generation sequencing of both MMC1 and MMC2 EVs to investigate their RNA profiles and compared against *C. albicans* EV-RNA. The analysis was performed by applying a minimum of 40% nucleotide similarity between the orthologues. The first observation is that the RNA compositions of MMC1 and MMC2 are very similar, and most of the molecules identified in *C. auris* EVs were noncoding RNAs (ncRNAs). The most represented ncRNAs in *C. auris* in both strains are fragments of tRNAs (Fig. 3A; see also Table S1 in the supplemental material). In *C. albicans*, the tRNAs are also the most represented ncRNAs and the snoRNAs are the second class of ncRNA in EVs; this enrichment is not observed in *C. auris* (Fig. 3B). For the tRNA fragments, there was an enrichment for the 3′ or 5′ end of the tRNAs; however, no enrichment in the central portion of the tRNA was observed (Fig. S1). Due to the library preparation, in the size selection step we cut the fragments related not only to the small fraction but also longer molecules, allowing the isolation of mRNAs as well. Transcriptome sequencing (RNA-seq) analysis led to the identification of 57 mRNAs in *C. auris* and 32 in *C. albicans*. The top 10 most abundant transcripts from each species are summarized in Table 1, and all of the mRNAs are listed in Table S2. The transcript antisense to rRNA, Tar1, was the most enriched in *C. albicans* EVs, followed by the mRNA

**TABLE 1** Gene ontology for RNAs enriched in EVs from *C. auris* and *C. albicans*[a]

### C. auris

| Name | Protein names | TPM mean | Log$_2$ fold change | FDR |
|---|---|---|---|---|
| CAALFM_C109220WA | MFS domain-containing protein | 1112.50 | 6.50 | 0.00% |
| CAALFM_C112570CA | Elongator subunit | 1357.63 | 6.63 | 0.00% |
| CAALFM_C207040WA | HIT-type domain-containing protein | 953.50 | 4.06 | 0.06% |
| POX18 | Pox18p | 809.32 | 6.06 | 0.00% |
| CAALFM_C111080WA | 6-phosphofructo-2-kinase | 494.49 | 2.87 | 1.39% |
| RBP1 | FK506-binding protein 1 (FKBP) (EC 5.2.1.8) (Peptidyl-prolyl cis-trans isomerase) (PPIase) (Rapamycin-binding protein) | 1586.55 | 8.36 | 0.00% |
| MET1 | Uroporphyrinogen-III C-methyltransferase | 1103.00 | 3.23 | 0.05% |
| STE24 | CAAX prenyl protease (EC 3.4.24.84) | 422.40 | 5.87 | 0.00% |
| RGS2 | GTPase-activating protein | 507.00 | 6.58 | 0.00% |
| CAALFM_C204870CA | THO complex subunit 2 | 238.36 | 5.07 | 0.00% |
| CAALFM_C114310WA | Deoxycytidine monophosphate deaminase | 842.16 | 7.38 | 0.00% |
| RGS2 | GTPase-activating protein | 507.00 | 6.58 | 0.00% |

### C. albicans

| Name | Protein names | TPM mean | Log$_2$ fold change | FDR p-value |
|---|---|---|---|---|
| TAR1 | Tar1p | 14740.21 | -14.32 | 0.00% |
| CDC42 | Cell division control protein 42 homolog | 2280.43 | -3.90 | 0.01% |
| THR4 | Threonine synthase (EC 4.2.3.1) | 1486.19 | -5.44 | 0.00% |
| THS1 | Threonyl-tRNA synthetase (EC 6.1.1.3) | 1624.40 | -5.15 | 0.00% |
| PHHB | 4a-hydroxytetrahydrobiopterin dehydratase (EC 4.2.1.96) | 1763.12 | -6.29 | 0.00% |
| CAALFM_CR02690WA | CAP-Gly domain-containing protein | 753.92 | -4.94 | 0.00% |
| CAALFM_C107850CA | Anaphase promoting complex subunit | 1198.57 | -3.16 | 0.26% |
| GCN20 | Putative AAA family ATPase | 1083.26 | -3.31 | 0.05% |
| CAALFM_C501410CA | ER membrane protein complex subunit 1 | 392.62 | -3.33 | 0.01% |
| POM152 | Pom152p | 390.75 | -3.55 | 0.00% |
| MRPL8 | Mitochondrial 54S ribosomal protein YmL8 | 559.14 | -4.39 | 0.00% |
| CAALFM_C100200CA | RRM domain-containing protein | 341.91 | -3.17 | 0.00% |

[a]TPM, transcripts per million; FDR, false-discovery rate.

coding cell division control protein 42 homolog and other transcripts related to the cell cycle (Table S2). For *C. auris*, the enriched transcripts were peptidyl-prolyl *cis-trans* isomerase, rapamycin-binding protein, translation elongation factor 1 subunit beta, E3 ubiquitin-activating protein, and MFS family membrane transporter (Table S2). To validate the presence of full-length mRNAs, we selected only transcripts with read coverage greater than 10×. It is possible to observe that we obtained reads mapping along the entire transcript, such as, for example, the Tar1 mRNA (Fig. S2). In addition, it is also possible to observe that the transcript is enriched only in *C. albicans*; no reads map the Tar1 transcript in *C. auris* MMC1 and MMC2 strains (Fig. S2). Overall, our results showed that *C. auris* EVs carry RNA, as demonstrated previously for other fungi (30–32).

**Proteomics analysis of extracellular vesicles.** We performed a liquid chromatography–tandem mass spectrometry (LC-MS/MS)-based proteomic analysis to compare the protein profiles across both species. We utilized a threshold of 40% homology at

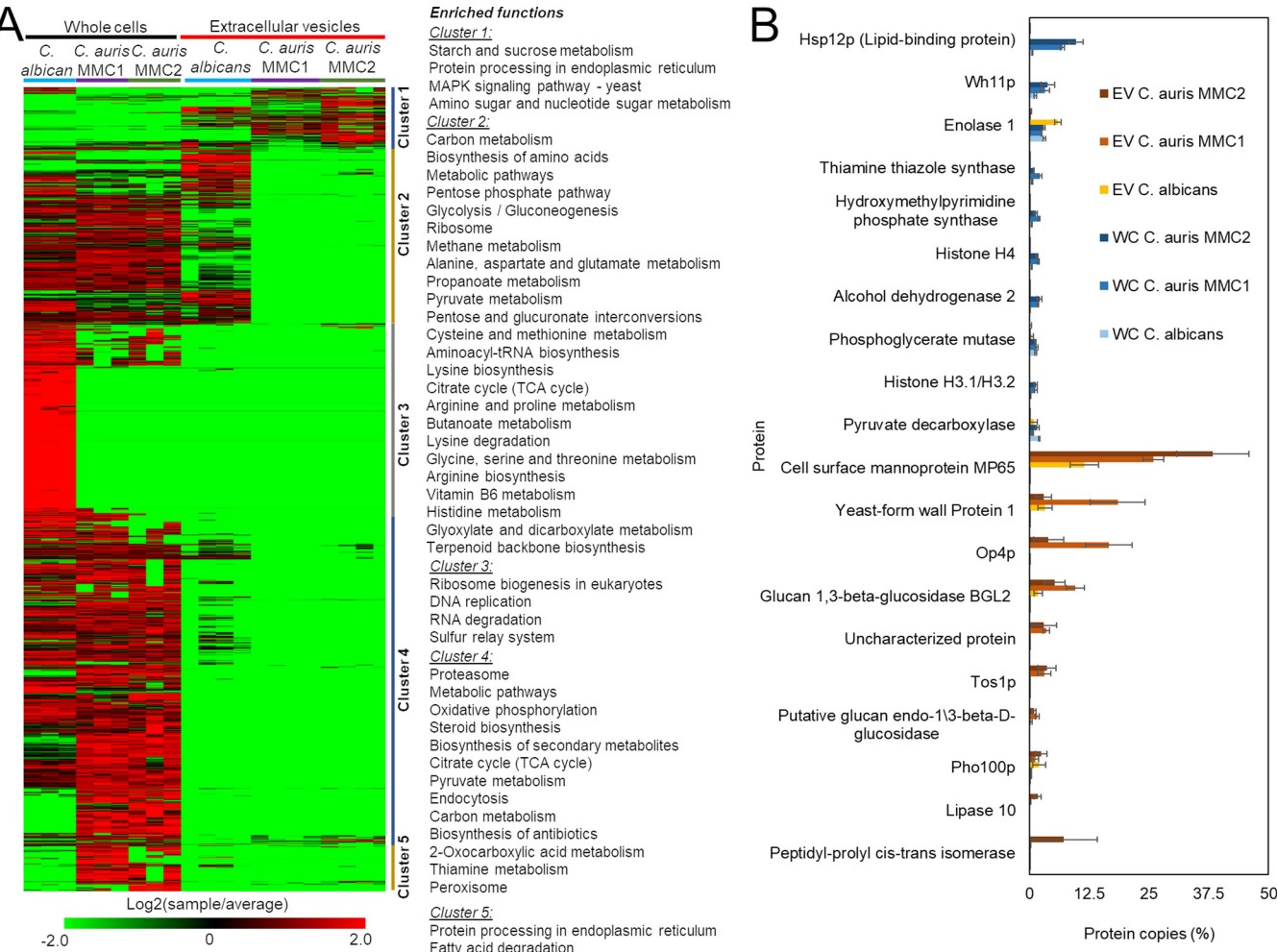

**FIG 4** Comparative proteomic analysis of extracellular vesicles (EVs) and whole cells from *C. albicans* and *C. auris*. Proteomic data of whole cells and extracellular vesicles were submitted to intensity-based absolute quantification (iBAQ) and converted to relative copy numbers before comparison across different samples. (A) Heatmaps and hierarchical clustering were performed with MeV and function enrichment of each cluster was performed with DAVID. (B) Profiles of the 10 most abundant whole-cell (WC) and extracellular vesicles (EVs) proteins. Cell samples correspond to three independent cultures and EVs samples correspond to 4 independent EVs isolations.

the amino acid level between the species to consider them homologs (Table S3). The divergent peptide sequences prevent direct comparison of the peak areas between the two species, since the sequence divergence might cause the peptides to not have same signal response in the mass spectrometer. Therefore, we calculated the relative copy number of proteins per sample (% from total) to compare between the two species. In addition to comparing the EV proteins from both species, we also compared the EV data with the proteomics analysis of whole cells (5), which were prepared and run in parallel. We observed striking differences between the whole cells and EVs for each of the 3 strains (Fig. 4A). We performed hierarchical clustering to separate groups of proteins based on their abundance profile. Cluster 1, which contains proteins commonly enriched in EVs from *C. auris* and *C. albicans* compared to those in their respective cells, were enriched in proteins from starch and sucrose metabolism, protein processing in the endoplasmic reticulum, MAP kinases, and amino sugar and nucleotide sugar metabolism (Fig. 4A). *C. albicans*, but not *C. auris*, EVs were enriched in abundant cellular proteins, such as ribosomal proteins and proteins from the central carbon and amino acid metabolisms (Fig. 4A, cluster 2). EVs from both species were depleted of proteins for functions such as ribosomal biogenesis, proteasome, DNA replication, RNA degradation, and sterol biosynthesis (Fig. 4A, clusters 3 to 5). Within the 10 most abundant proteins in whole cells, only enolase 1 and pyruvate decarboxylase had large

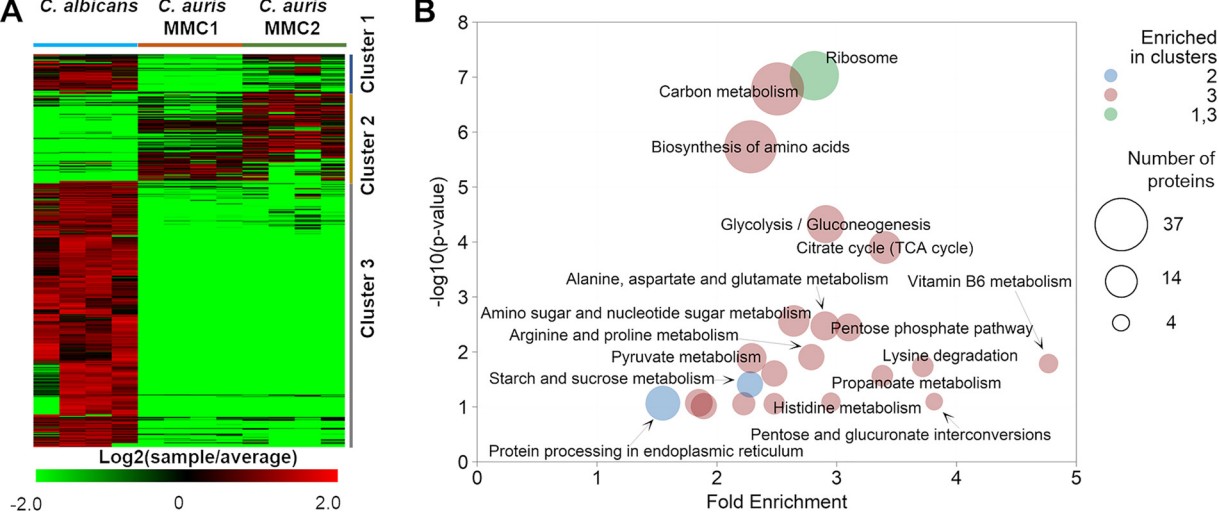

**FIG 5** Proteomics analysis of EVs from *C. auris*. EVs from both *Candida* species were submitted to protein extraction and analysis. (A) Heatmap showing the abundance of proteins differentially abundant in EVs from both species of *Candida*. The heatmap was clustered using the hierarchical method. (B) Function enrichment analysis of different clusters of proteins shown in panel A. Enrichment of pathways was done with DAVID. The graph represents the relationship between *P* values and fold enrichment. The colors of the circles represent the different clusters they are enriched in, while the sizes represent the numbers of proteins from each pathway. EV samples correspond to 4 independent EV isolations.

amounts in EVs (*C. albicans*) (Fig. 4B). None of the top 10 most abundant whole-cell proteins were abundant in *C. auris* EVs (Fig. 4B). On the other hand, the top 10 most abundant EVs proteins were present only in small amounts in whole cells (Fig. 4B), suggesting a highly selective process to upload proteins into EVs.

Among all the proteins detected in EVs, 393 were considered differentially abundant when *C. auris* (both strains) was compared with *C. albicans*. We performed hierarchical clustering of the differential abundant proteins, followed by functional enrichment analysis. To provide information on the number of proteins in each enrichment, we showed the pathway enrichment results as bubble graphs. In this layout, the enrichment fold change and *P* values are plotted on the *x* and *y*, axes, respectively, while the circle sizes represent the number of proteins and the colors represent the clusters they belong to. The number of differentially abundant proteins corresponded to 33% of the detected proteins, so the abundance of the remaining 66% was similar among the species. The heatmap in Fig. 5A shows all of the major differentially abundant EV proteins among the evaluated organisms. The heatmap was divided into three clusters based on differences in protein abundance between EVs from *C. auris* and *C. albicans*. Out of the 393 proteins on the heatmap, 42 proteins (~10%) were more abundant in *C. auris* than in *C. albicans* (Fig. 5, cluster 2). This group of proteins was enriched in proteins from starch and sucrose metabolism and protein processing in the endoplasmic reticulum (Fig. 5B). As mentioned above, *C. albicans* EVs had larger amounts of metabolic proteins (Fig. 5A and B, cluster 3). *C. albicans* EVs had larger amounts of tricarboxylic acid (TCA) cycle proteins (Fig. 5B), which is the opposite of what was found in whole cells (5). This result further supports the presence of a selective mechanism for sorting proteins into the EVs.

**Lipidomics analysis of extracellular vesicles.** We performed a lipidomics analysis to compare the lipid profiles of *C. auris* and *C. albicans* EVs (Fig. 6A and B). All detected species of diacylglycerols (DG) and triacylglycerols (TG) were more abundant in EVs from *C. albicans*, whereas the majority of glycerophospholipids were enriched in the *C. auris* isolates, including phosphatidylcholine (PC), phosphatidylethanolamine (PE), phosphatidylglycerol (PG), and phosphatidylserine (PS) species (Fig. 6A). Conversely, phosphatidic acid (PA) and phosphatidylinositol (PI) species were more abundant in *C. albicans* (Fig. 6A). The pattern of sphingolipids was also

**mSystems**®

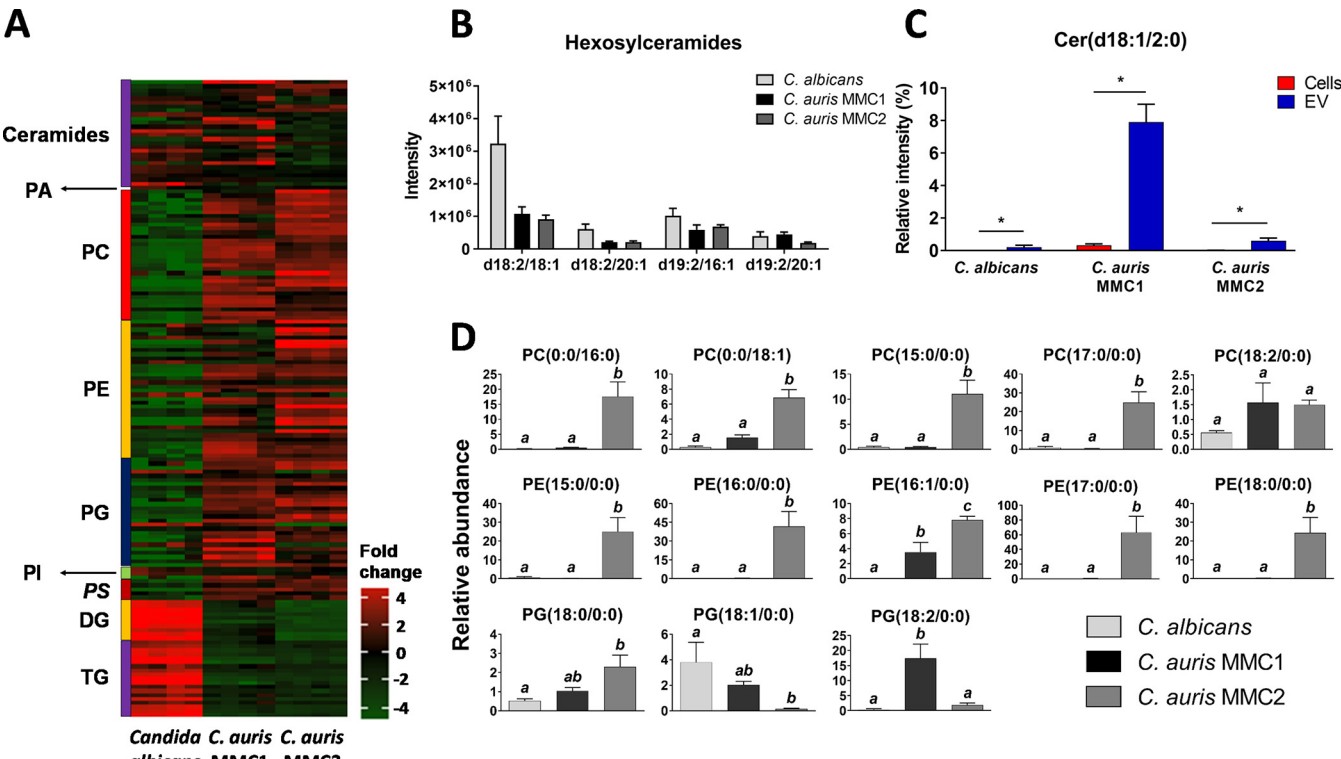

**FIG 6** Lipid profile of *C. auris* and *C. albicans* EVs. Vesicles from both *Candida* species were submitted to lipid extraction and analysis by liquid chromatography-tandem mass spectrometry (LC-MS/MS). (A) Heatmap of the relative abundances of EV lipids from both *Candida* species. (B) Relative intensity of hexosylceramides. (C) Relative intensity of Cer(d18:1/2:0) was compared between EVs and yeast cells among all *Candida* species. (D) Relative abundance of lysophospholipids in *C. auris* and *C. albicans* EVs. Data were analyzed by one-way ANOVA followed by Tukey's multiple-comparison test. Within each graph, different letters above the bars indicate $P < 0.05$, and same letters indicate $P > 0.05$. EV samples correspond to 4 independent EV isolations. Cer, ceramide; DG, diacylglycerol; PA, phosphatidic acid; PC, phosphatidylcholine; PE, phosphatidylethanolamine; PG, phosphatidylglycerol; PI, phosphatidylinositol; PS, phosphatidylserine; TG, triacylglycerol.

distinct when the *Candida* EVs were compared. Two major species of conserved hexosyl-ceramides (HexCer) were found in EVs from *C. albicans* and *C. auris* (Fig. 6B), correspond-ing to the same distribution characterized in their respective yeast extracts that was recently reported by our group (5). Consistent with that, HexCer species bearing Cer (d18:1/24:0(2OH)) and Cer(d20:0/18:0) were more abundant in *C. albicans* EVs. The nona-cylated sphingoid bases sphinganine Cer(d18:0/0:0) and sphingosine Cer(d18:1/0:0) were more abundant in *C. auris* MMC2 (Table S4). We also found unusual free ceramide spe-cies with acetate as the acyl group (Fig. 6C), known as C2 ceramides, which were more abundant in *C. auris* MMC1. Remarkably, Cer(d18:1/2:0) comprises 7.9% of the mass spec-trometry signal for all identified lipids in the positive-ion-mode analysis of *C. auris* MMC1 EVs, but only 0.6% and 0.2% of the signal for *C. auris* MMC2 and *C. albicans* EVs, respec-tively (Fig. 6C). We compared the relative intensities of Cer(d18:1/2:0) to the whole-cell data from our recent publication (5). We found 25-, 67-, and 109-fold enrichments of this lipid species in EVs compared to the levels in whole cells in *C. auris* MMC1, *C. auris* MMC2, and *C. albicans*, respectively.

Recently, we reported that *C. auris* had higher expression than *C. albicans* of a vari-ety of phospholipases (5); therefore, we took a closer look at their products, lysophos-pholipids (33). *C. auris* MMC2 EVs had consistently higher abundances of lysophospha-tidylcholine (LPC) and lysophosphatidylethanolamine (LPE) species than those in *C. auris* MMC1 and *C. albicans* (Fig. 6D). The lysophosphatidylglycerol (LPG) species PG (18:0/0:0), PG(18:1/0:0), and PG(18:2/0:0) were more abundant in *C. auris* MMC2, *C. albi-cans*, *C. auris* MMC1, respectively (Fig. 6D).

Fatty-acid (FA) chains were detected in all organisms and ranged in size from 14 to 24 carbons, and arachidonic acid was consistently detected, esterified to

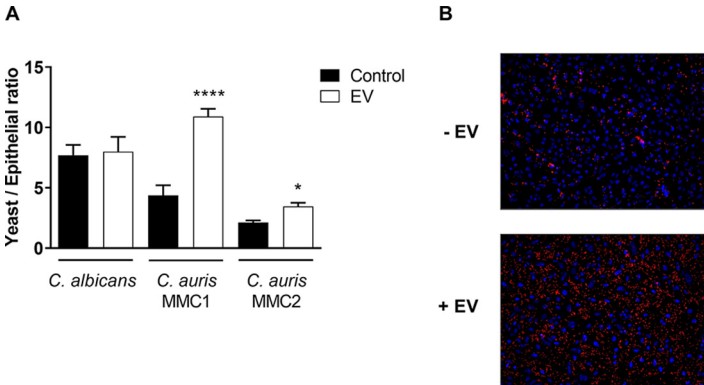

**FIG 7** The effect of extracellular vesicle (EV) pretreatment on the adhesion of yeast cells to epithelial monolayers. Epithelial cells were pretreated or not with EVs (10 μg/ml of protein) for 1 h before challenge with the respective yeast cells. After incubation for 1 h, monolayers were washed, and slides were analyzed under a fluorescence microscope (see Materials and Methods for details). (A) Quantification of adhering yeast cells. (B) Fluorescence image of *C. auris* MMC1 adhesion to epithelial cells. Nuclei were stained with 4′,6-diamidino-2-phenylindole (DAPI) (blue), whereas the yeasts were stained with *N*-hydroxysuccinimide (NHS)-rhodamine (red). The graph shows average and standard errors relative to 2 independent experiments made with distinct EV preparations. *, $P < 0.05$ by two-tailed paired *t* test.

phosphatidylethanolamine [PE(18:2/20:4)], in both *C. auris* isolates. Arachidonic acid was not abundant in the evaluated strain of *C. albicans* under the tested conditions (Table S4).

**Effects of extracellular vesicles on yeast adhesion to epithelial cells.** Adhesion to epithelial surfaces is an important feature displayed by pathogenic species of *Candida* as an early stage of colonization of host tissues (34–37). We evaluated whether *C. albicans* or *C. auris* EVs had an impact on the adhesion of *C. auris* or *C. albicans* to epithelial monolayers. Preincubation of epithelial cells with *C. auris* EVs increased the adhesion of the yeast cells to the monolayers. This did not occur with *C. albicans*, as the pretreatment with their EVs did not affect the adhesion of yeast cells to epithelial cells (Fig. 7). This result shows how EVs from *C. auris* can induce distinct yeast-host cell phenotypes compared to those of *C. albicans*.

**Effect of extracellular vesicles on phagocytosis and killing by macrophages.** EVs from certain fungi can affect the way yeast cells are internalized and killed by macrophages (8, 13–16, 38). We tested whether EVs from *C. auris* or *C. albicans* would be able to modulate the uptake or clearance of yeast cells by macrophages. Incubation with EVs from either *C. albicans* or *C. auris* had no significant effect on the phagocytosis of yeast cells by macrophages (Fig. 8A). However, whereas EV preincubation enhanced

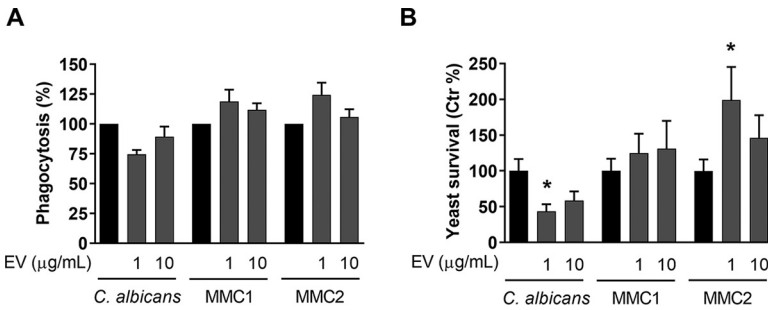

**FIG 8** Phagocytosis and killing by macrophages. (A) RAW 264.7 macrophages were incubated with EVs for 1 h and then challenged with yeast cells in the ratio of 1:2 (macrophage-yeast) for 1 h. After this period, extracellular yeast cells were washed off, and macrophages were lysed and plated onto Sabouraud agar for CFU counting. (B) Bone marrow-derived macrophages were incubated with EVs for 4 h and then challenged with yeast in a ratio of 10:1 (macrophages-yeast) for 24 h. The macrophages were then lysed and plated onto Sabouraud agar for CFU counting. Graphs show averages and standard error of the mean for 4 independent experiments made with distinct EV preparations. *, $P < 0.05$ by two-tailed paired *t* test.

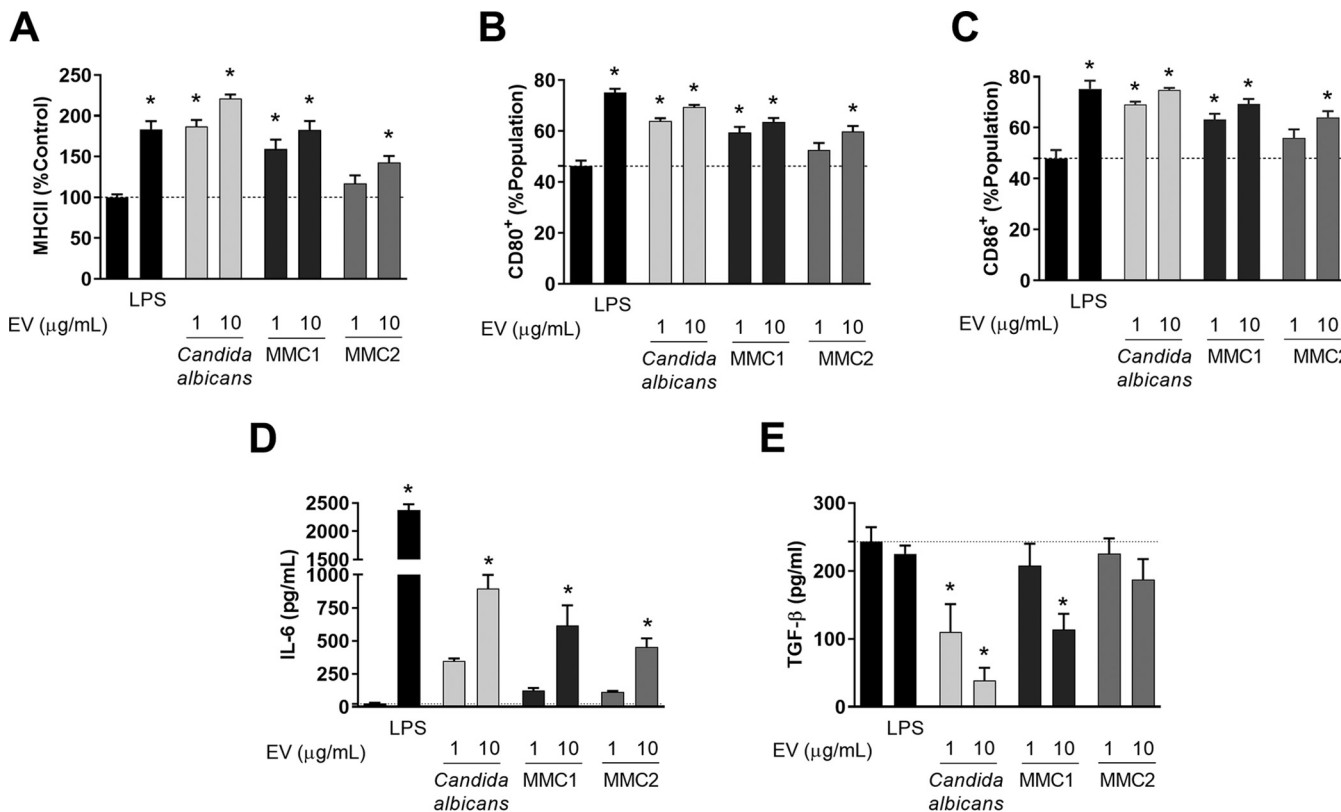

**FIG 9** Activation of bone marrow-derived dendritic cells (BMDCs) by *C. auris* extracellular vesicles (EVs). BMDCs were incubated or not with EVs from *C. albicans* and *C. auris* for 24 h and then analyzed by flow cytometry (A to C) for the expression of MHC-II (A), CD80 (B), and CD86, or by enzyme-linked immunosorbent assay (ELISA) (D and 7E), for the production of interleukin-6 (IL-6) (D) and transforming growth factor beta (TGF-$\beta$) (E). Graphs show mean and standard error for 3 independent experiments made with distinct EV preparations. *, $P < 0.05$ by one-way ANOVA followed by Tukey's multiple-comparison test.

the ability of macrophages to kill *C. albicans*, EVs from *C. auris* MMC2, but not from MMC1, enhanced yeast cell proliferation within the macrophages (Fig. 8B). This points to differing roles for EVs among species regarding some of the effector functions of macrophages.

**Activation of dendritic cells by extracellular vesicles.** We investigated the ability of *C. auris* to regulate dendritic cells by measuring 3 important signals for antigen presentation and activation of T cells, MHC-II, costimulatory molecules (CD80 and CD86), and cytokines (39). Bone marrow-derived dendritic cells (BMDCs) were incubated for 24 h with *C. auris* or *C. albicans* EVs, and MHC-II and costimulatory molecules were measured by fluorescence-activated cell sorting (FACS), whereas cytokines were assayed by enzyme-linked immunosorbent assay (ELISA). We observed an increase of surface markers associated with BMDC activation, which was similar to that induced by lipopolysaccharide. Despite that, MMC2 EVs induced a lesser response, and all tested EV concentrations from *C. auris* and *C. albicans* were able to increase, in a dose-dependent manner, the expression of MHCII, CD80, and CD86 on BMDCs (Fig. 9A to C). BMDCs treated with EVs from both *Candida* species did not produce IL-10, IL-12, or TNF-$\alpha$. However, BMDCs stimulated with EVs from *C. auris* produced IL-6 similarly to *C. albicans* (Fig. 9D). In addition, inhibition of the basal production of transforming growth factor beta (TGF-$\beta$) by BMDCs was detected after the incubation with EVs from *C. auris* MMC1 and *C. albicans* (Fig. 9E).

## DISCUSSION

The literature reports two distinct populations of EVs from other species of *Candida*, a smaller one, usually ranging from 50 to 70 nm, and a larger group with a size between 100 and 800 nm (9, 24, 29). We found that *C. auris* releases EVs encompassed by lipid bilayers with size and shape consistent with those from other species. TEM

showed that, as opposed to EVs of some species, such as *Histoplasma capsulatum* (24) and *Cryptococcus neoformans* (23), *C. auris* EVs lack electron-dense areas associated with pigmentation. The total contents of protein and ergosterol in EV suspensions were higher in *C. albicans* than in either *C. auris* strain when normalized by the number of producing cells. The ratio between the concentrations of protein and sterol is a way to measure the main components present in EVs, regardless of the number of EV-producing cells, and *C. auris* MMC2 had a ratio higher than those of *C. albicans* or *C. auris* MMC1.

EV RNAs have been characterized in *C. neoformans*, *Saccharomyces cerevisiae*, *Paracoccidioides brasiliensis*, *H. capsulatum*, and *C. albicans* (30, 31). The most abundant transcripts of EVs from *C. auris* MMC1 and MMC2 were associated with general metabolism, RNase, and ubiquitin activities or were from uncharacterized genes. EV mRNAs of *C. albicans* and *C. auris* share common biological processes, such as cellular response to stress and filamentous growth, indicating a conserved sorting mechanism (30). In other eukaryotes, EV mRNAs can be translated into the recipient cell (40), although this should be experimentally addressed for fungal EVs. The most abundant ncRNAs in EVs from *C. auris* MMC1 and MMC2 were tRNAs and their fragments, similarly to what was previously described for *C. albicans* ($\sim$60%). These fragments of tRNA have been described in EVs of diverse organisms, from unicellular parasites to human cells (41–43). In T lymphocyte-derived EVs, the most abundant class of RNA characterized is tRNA fragments, which comprise 45% of all RNA identified in the EVs compared to the cell content. These fragments act by repressing immune activation in T cells (44).

The proteomic profile of EVs was strikingly different from that of the cells they are derived from. Whereas *C. auris* EVs were enriched in proteins from starch and sucrose metabolism and protein processing in the endoplasmic reticulum, *C. albicans* EVs had larger amounts of proteins from central carbon metabolism, ribosomes, and amino acid metabolism. We have previously shown that TCA cycle proteins were more abundant in *C. auris* yeast cells than in *C. albicans* (5), but we are now showing an opposite phenotype in EVs, suggesting a selective sorting that could help control the intracellular levels of specific metabolic enzymes. These differences suggest that the EVs from *C. albicans* and *C. auris* are involved with distinct metabolic adaptations. In terms of lipid composition, the relative abundances of lipids involved with energy storage, as triacyl- and diacylglycerols (TG and DG), are remarkably higher in EVs from *C. albicans* than in those from *C. auris* EVs, reflecting the pattern found in their originating yeast cells (5). The relative abundance of structural glycerophospholipids is consistently higher in EVs from both isolates of *C. auris*, also reflecting the lipid profile of their generating yeast cells (5). Although in some cases the lipid profile from *C. auris* EVs resembled the yeast cell one, some lipids from the yeast cells were not present in EVs, including cardiolipins, which are mitochondrial markers. The amount of HexCer correlated with the distribution in their respective cells, as previously shown by our group (5). Considered initially as membrane structural components, HexCer were described as virulence regulators in *C. albicans* and *C. neoformans* (45, 46). Their role in EVs could be linked to membrane and lipid raft stability (47). However, recently, Xisto and colleagues demonstrated that purified HexCer produced by the opportunistic fungus *Lomentospora prolificans* induced an oxidative burst by and increased the antifungal activity of macrophages (48).

To our knowledge, our findings report the first time that a C2 ceramide derivative has been found in fungal EVs. In mammalian models, C2 ceramide has biological properties, such as antitumoral activity, that induce apoptosis and arrest the cell cycle (49, 50). The relative abundance of lysophospholipids was considerably higher in EVs from *C. auris* than in those from *C. albicans*, particularly in EVs from MMC2. These data suggest an intense activity in *C. auris* of lipid catabolic enzymes, such as phospholipases. Some lysophospholipids are biologically active on leukocytes; for instance, LPC released by apoptotic neutrophils recruits monocytes from the bloodstream to promote clearance of apoptotic bodies from tissues (51). Immunomodulatory properties

of LPC have been demonstrated for other infection models, and LPC could act as a virulence factor in *C. auris* infections (52, 53).

To examine the potential biological effects of EVs upon host cells and to compare biological activities between species, we used amounts of EVs based on protein concentration in the same range used in other reports for host-pathogen studies (8, 9, 16, 17). Within this concentration range, EVs from other pathogenic fungi were proven to be biologically active in distinct models. Adhesion of yeast cells to epithelial surfaces is an important mechanism of disease as an initial step for further tissue damage and colonization of distinct sites in the host, including the bloodstream (37). *C. albicans* can interact with surface adhesion molecules on epithelial cells (35, 36), so we investigated whether EVs from both *Candida* species were able to modulate the adhesion of yeast to epithelial monolayers *in vitro*. Increased yeast cell adhesion to the monolayers was observed when EVs from *C. auris* (from either strain) were added to the epithelial cells, but this did not occur when *C. albicans* EVs and yeast cells were examined. Notably, *C. neoformans* EVs fuse with brain microvascular endothelial cells, changing their permeability (21). Since in our study, the epithelial cells were incubated with EVs prior to the challenge with yeast cells, it is possible that fusion with the epithelial cells could modify their permeability and/or modulate the exposure of adhesion molecules, although further experimentation is needed to address this hypothesis. Molecules involved with the adhesion of *C. albicans* to epithelial cells, such as *C. albicans* ALs3p and Eap1p, have been described previously and are potential players in the increase of adhesion induced by EVs (37).

Fungal EVs can induce the activation of phagocytes, increasing phagocytosis, cytokine production, and antigen presentation (8–10, 17). EVs isolated from both *C. auris* strains did not modulate the uptake of yeast cells by macrophage cell lines such as RAW, but EVs from *C. auris* MMC2 inhibited the killing of yeast cells by BMDMs. EVs from *C. albicans* increased the killing of yeast cells by BMDMs. EVs from other pathogenic fungi can modulate phagocytosis and/or killing by macrophages, but our data show that EVs from only one of the *C. auris* isolates (MMC2) could inhibit the killing of the pathogen by macrophages. These data suggest that EVs from the same species could promote distinct changes in host cells. EVs from *C. albicans* followed the pattern exhibited by most fungal EVs, as they induced killing (8, 10, 14–16, 38). Incubation with EVs stimulated BMDCs to express important signals responsible for CD4$^+$ T-cell activation, such as MHCII, CD80, and CD86. The secretion of TNF-$\alpha$, IL-10, and IL-12p70 by BMDCs was not detected at biologically relevant levels. However, EVs from both *C. albicans* and *C. auris* induced the release of IL-6 by BMDC while decreasing the basal production of TGF-$\beta$. This suggests that EVs from *C. albicans* and *C. auris* MMC1 induce an inflammatory response in BMDCs. Unlike in a previous report (9), TNF-$\alpha$, IL-10, and IL-12p70 were not produced by BMDCs stimulated with EVs. Different strain of *C. albicans* were used in these studies, reinforcing the possibility that the biological activity of EVs could be strain specific.

In summary, our results show that the emerging pathogen *C. auris* produces EVs that are similar in size to those from other pathogenic fungi, but the content of these EVs distinctly differs from what is known for *C. albicans*, and these differences could explain the phenotypic changes induced by these EVs in the cells from the host. In this regard, we note that *C. auris* is a new fungal pathogen that has been proposed to have emerged from the environment as a result of global warming (54). In contrast, *C. albicans* has an ancient association with human hosts. Thus, the similarities in structure and content between *C. auris* and *C. albicans* EVs probably reflect constraints common to fungal cells and their physiology, while the differences reflect species-specific variables and perhaps the result of differences in the time of adaptation to human hosts.

## MATERIALS AND METHODS

**Cell lines.** Two well-characterized *C. auris* clinical isolates (MMC1 and MMC2) were acquired from Montefiore Medical Center (New York, USA) (5). *C. albicans* strain ATCC 90028, RAW 264.7 macrophages (ATCC TIB-71), and HeLa cells (ATCC CCL-2) were obtained from ATCC. Yeast cells were cultivated in

yeast-peptone-dextrose (YPD) broth and seeded onto Sabouraud agar plates. For each experiment, colonies were inoculated in Sabouraud broth for 24 h at 30°C before use. RAW 264.7 and HeLa cell lines were cultivated up to the 10th passage in Dulbecco's modified Eagle's medium (DMEM) supplemented with 10% fetal bovine serum (FBS) and 1% nonessential amino acids.

**EV isolation.** One colony of each strain of *C. auris* or *C. albicans* was inoculated in 10 ml of Sabouraud broth for 24 h at 30°C and then expanded in 200 ml of fresh medium. After an additional 24 h at 37°C, the cells were centrifuged. The supernatant was filtered and concentrated 40-fold using an Amicon system with a 100-kDa molecular-weight-cutoff membrane. The concentrate was centrifuged twice at 150,000 × *g* at 4°C for 1 h, with a phosphate-buffered saline (PBS) washing step between each centrifugation step. The EV pellets were suspended in filtered PBS for most of the experiments and in 50 mM ammonium bicarbonate for proteomic and lipidomic analyses.

**Transmission electron microscopy.** EV pellets were fixed in 2.5% glutaraldehyde and 3 mM $MgCl_2$ in 0.1 M sodium cacodylate buffer (pH 7.2) overnight at 4°C. Samples were then rinsed with buffer and postfixed in 0.8% potassium ferrocyanide-reduced 1% osmium tetroxide in the buffer for 1 h on ice in the dark. After a 0.1 M sodium cacodylate buffer rinse, the samples were incubated at 4°C overnight in the same buffer. Samples were rinsed with 0.1 M maleate buffer, *en bloc* stained with 2% uranyl acetate (0.22 $\mu$m filtered, 1 h, dark) in 0.1 M maleate, dehydrated in a graded series of ethanol, and embedded in Eponate 12 (Ted Pella) resin. Samples were polymerized at 37°C for 2 days and at 60°C overnight. Thin sections (60 to 90 nm) were cut with a diamond knife on a Reichert-Jung Ultracut E ultramicrotome and picked up with Formvar-coated copper slot grids. Grids were stained with 2% uranyl acetate in 50% methanol, followed by lead citrate, and observed with a Phillips CM120 transmission electron microscope at 80 kV. Images were captured with an XR80 high-resolution (16-bit) 8-megapixel camera (AMT).

**Protein and ergosterol quantification.** Protein and sterols were quantified using bicinchoninic acid (BCA) protein assay (Thermo) and Amplex red cholesterol assay (Thermo) kits, respectively. Both contents were expressed as a function of the number of yeast cells present in each culture at harvest time.

**Hydrodynamic size distribution of extracellular vesicles by dynamic light scattering.** EVs were suspended in PBS, and their hydrodynamic size distributions were measured in a BI-90 Plus particle size analyzer (Brookhaven Instruments) at room temperature as described previously (55). Vesicle preparations were first centrifuged at 13,000 rpm for 5 min to remove any larger particles and aggregates. One hundred microliters of sample were loaded into a disposable cuvette (catalog no. 952010077; Eppendorf) and analyzed by dynamic light scattering (DLS). The average size distribution was calculated from duplicates of 10 individual measurements.

**Isolation and sequencing of extracellular vesicles RNAs.** The RNA molecules were isolated with the miRNeasy minikit (Qiagen), which enables the purification of molecules from 18 nucleotides (nt) up to m RNAs, according to the manufacturer's protocol. This allowed us to obtain not only the small-RNA-enriched fractions but also molecules longer than >200 nt. The RNA profile was assessed in a 2100 Bioanalyzer (Agilent Technologies). The purified RNA, from three independent biological replicates, was used for RNA-seq library construction using the TruSeq small RNA kit (Illumina) according to the manufacturer's recommendations with a slight modification. During the acrylamide gel size selection, we excised the band ranging from 18 nt to >200 nt in length. The sequencing was performed with the Illumina HiSeq 2500 platform and the TruSeq SBS kit v3-HS 50-cycle kit (Illumina).

***In silico* data analysis.** The RNA-seq analysis was performed with CLC Genomics Workbench software v20. The *C. auris* B8441 genome used for mapping was obtained from the NCBI database (GenBank accession number GCA_002759435.2). The alignment was performed as follows: additional 100-base upstream and downstream sequences, minimum number of reads = 10, maximum number of mismatches = 2, nonspecific match limit = −2, and minimum fraction length of 0.8 for the RNA mapping. The minimum read similarity mapped on the reference genome was 80%. Only uniquely mapped reads were considered in the analysis. The libraries were normalized per million, and the expression values for the transcripts were registered in transcripts per million (TPM). For the ncRNA, the database used was the ncRNA from the *Candida* genome database: C_auris_B8441_version_sXX-mYY-rZZ_other_features_no_introns.fasta.gz. For the mRNA identification in the EVs, we combined the differential expression with reads coverage, so we mapped reads to reference (C_auris_B8441_version_s01-m01-r10_genomic and C_auris_B8441_version_s01-m01-r10_other_features_plus_intergenic) using the following parameters: no masking, match score = 1, mismatch cost = 2, linear insertion cost = 3) deletion cost = 3, length fraction = 0.6, similarity fraction = 0.8, and global alignment. To consider the full-length mRNAs, we selected those with both expression values (TPM) greater than 100 and 5× transcript coverage. Gene ontology analysis was performed using the DAVID annotation tool (56).

**Lipidomics and proteomics analyses of extracellular vesicles.** Sample processing and analysis were carried out as described previously (57, 58). Briefly, samples were submitted to simultaneous metabolite, protein, and lipid extraction (MPLEx) (57). Extracted lipids were dried in a vacuum centrifuge and dissolved in methanol before analysis by liquid chromatography-tandem mass spectrometry (LC-MS/MS) on a Velos Orbitrap mass spectrometer (Thermo Fisher). Lipid species were identified and manually inspected for validation based on head group and fatty acyl chain fragments using LIQUID (59). The intensity of each lipid species was extracted using MZmine v2.0 (60).

Proteins were dissolved in 100 $\mu$l of 50 mM $NH_4HCO_3$ containing 5 mM dithiothreitol and 8 M urea, and incubated for 15 min at 37°C. Reduced thiol groups were alkylated with a final concentration of 10 mM iodoacetamide (from a 400 mM stock solution) incubated for 30 min at room temperature. The reaction was quenched by adding 500 mM dithiothreitol to a final concentration of 20 mM. Samples were diluted 8-fold with 50 mM $NH_4HCO_3$ containing 1 mM $CaCl_2$ and digested with overnight at 37°C with 2 $\mu$g of sequencing-grade trypsin (Promega). Samples were desalted with solid-phase extraction

$C_{18}$ spin columns (UltraMicro spin columns, 3- to 30-$\mu$g capacity; Nest Group) as previously described (27).

The resulting peptides were dissolved in water and loaded into a $C_{18}$ trap column (4-cm by 100-$\mu$m inner diameter [ID], packed in-house with 5-$\mu$m $C_{18}$; Jupiter). Chromatography was carried out on a capillary column (70-cm by 75-$\mu$m ID packed with $C_{18}$, 3-$\mu$m particles) using a gradient of acetonitrile (mobile phase B) in water (mobile phase A), both supplemented with 0.1% formic acid. The elution was carried out at 300 nl/min with the following gradient: 19 min, 8% B; 60 min, 12% B; 155 min, 35% B; 203 min, 60% B; 210 min, 75% B; 215 min, 95% B; 220 min, 95% B. Eluting samples were analyzed online with a Q-Exactive Plus mass spectrometer (Thermo Fisher Scientific). Full-scan spectra were collected in a window of 400 to 2,000 $m/z$ with a resolution of 70,000 at $m/z$ 400. The 12 most intense parent ions were submitted to high-energy collision dissociation (32% normalized collision energy) at a resolution of 17,500. Dynamic exclusion was set to fragment each parent ion once, excluding them for 45 sec.

Peptides were identified using MaxQuant v1.5.5.1 (61) by searching against the *C. albicans* SC5314 and *C. auris* sequences from UniProt Knowledge Base (downloaded 6 December 6 2017). The intensity-based absolute quantification (iBAQ) method was used for quantification (62). The iBAQ values for individual proteins were normalized against the total sum of all proteins, resulting in the relative protein copy number (percentage from total). *C. auris* and *C. albicans* proteins were considered orthologous with $\geq$40% amino acid sequence similarity (5). Heatmapping and clustering were performed with MultiExperiment Viewer (MeV) (63) or with R software and the Complex Heatmap package (64). For calculating fold changes and plotting the heatmaps, missing values were filled with half of the minimum value of the data set. Function enrichment analysis was done with DAVID (65) using default parameters. Bubble graphs were plotted using Minitab v19.2020.1. The mass spectrometry proteomics data have been deposited in the ProteomeXchange Consortium via the PRIDE (66) partner repository with the data set identifiers PXD026767 and PXD026768.

**Adhesion assay to epithelial monolayers.** HeLa cells were seeded on coverslips placed in 24-well plates and incubated for 24 h at 37°C. Cell monolayers were preincubated with *C. auris* or *C. albicans* EVs (10 $\mu$g/ml of protein) for 1 h and challenged with respective yeasts (prestained with *N*-hydroxysuccinimide [NHS]-rhodamine for 30 min at 30°C under shaking) for 1 h at a ratio of 20 yeast cells per HeLa cell. NHS-rhodamine staining does not change yeast cell growth rates or other cellular characteristics (data not shown). After extensive washing with PBS to remove nonadherent yeast, the cells were fixed with formalin and mounted with mounting medium containing 4′,6-diamidino-2-phenylindole (DAPI). Images were taken using a fluorescence microscope (Imager Z1; Zeiss), and the adhesion was measured by the ratio between NHS–rhodamine-positive cells divided by DAPI-positive cells for each field, using ImageJ. At least 8 fields containing approximately 400 epithelial cells per field from each slide were counted.

**Analysis of bone marrow-derived dendritic cell activation by extracellular vesicles.** Bone marrow-derived dendritic cells (BMDCs) were differentiated as described previously (67). Briefly, bone marrow cells were isolated from male C57BL/6 mice (approved protocol 2014-0501) by flushing both tibias and femurs with RPMI medium supplemented with 10% fetal bovine serum (FBS). Bone marrow cells were cultivated for 10 days at 37°C in the presence of granulocyte-macrophage colony-stimulating factor (GM-CSF) (Peprotech) (67). Cultures were fed with medium containing GM-CSF at days 3, 6, and 8. The BMDC phenotype was evaluated on day 10 by the surface exposure of CD11c and MHCII. At day 10 of differentiation, BMDCs were incubated with 1 and 10 $\mu$g/ml (protein) of EVs from *C. auris* and *C. albicans* for 24 h at 37°C and 5% $CO_2$. After this period, cytokines IL-6, IL-10, IL-12p70, TNF-$\alpha$, and TGF-$\beta$ were measured in the culture supernatants using ELISA. BMDCs were labeled with antibodies ($\alpha$-CD11c, $\alpha$-MHCII, $\alpha$-CD80, and $\alpha$-CD86) to evaluate their purity and activation state using flow cytometry.

**Modulation of effector functions of macrophages by extracellular vesicles. (i) Phagocytosis.** RAW 264.7 macrophages were plated onto 96-well plates and incubated for 24 h at 37°C. Cells were then incubated with EVs from *C. albicans* or *C. auris* (10 $\mu$g/ml of protein) for 1 h until challenge with the respective yeast cell at a ratio of 1:2 (macrophage-yeast) for 1 h. Plates were washed to remove extracellular yeast cells and then lysed with sterile water for CFU analysis.

**(ii) Killing.** Bone marrow cells were harvested from C57BL/6 mice as detailed above and incubated with RPMI medium containing 10% fetal bovine serum and 20% L929 supernatant at 37°C. On the fourth day, new medium containing L929 supernatant was added to the culture. On the seventh day of culture, the cells had matured to differentiated macrophages, confirmed by the expression of F4/80 and absence of LY6C. BMDMs were plated in 96-well plates and incubated at 37°C for 24 h. Cells were incubated with EVs for 4 h at 37°C until the challenge with yeast cells at 10:1 (macrophage-yeast) for 24 h at 37°C. Cells were lysed and the suspensions plated onto Sabouraud plates for CFU counting.

**Statistical analyses.** All experiments were performed at least 3 independent times, unless stated otherwise. Data sets were analyzed using one-way analysis of variance (ANOVA) and Dunnett's or Tukey's multiple-comparison posttest using Prism 8 (GraphPad Software, Inc.). All $P$ values of $<0.05$ were considered significant.

## SUPPLEMENTAL MATERIAL

Supplemental material is available online only.

**FIG S1**, TIF file, 0.3 MB.

**FIG S2**, TIF file, 0.1 MB.

**TABLE S1**, XLS file, 2.8 MB.

**TABLE S2**, XLS file, 3.7 MB.
**TABLE S3**, XLSX file, 0.2 MB.
**TABLE S4**, XLSX file, 0.05 MB.

## ACKNOWLEDGMENTS

D.Z.M., J.D.N., and E.S.N. were supported by National Institutes of Health (NIH)–National Institute of Allergy and Infectious Diseases grant R21 AI124797. L.N. was supported by grants from the Brazilian agency Conselho Nacional de Desenvolvimento Científico e Tecnológico (CNPq; grants 311179/2017-7 and 408711/2017-7) and FAPERJ (E-26/202.809/2018). M.L.R. was supported by grants from the Brazilian Ministry of Health (grant 440015/2018-9), Conselho Nacional de Desenvolvimento Científico e Tecnológico (grants 405520/2018-2 and 301304/2017-3), and Fiocruz (grants PROEP-ICC 442186/2019-3, VPPCB-007-FIO-18, and VPPIS-001-FIO18). M.L.R. also acknowledges support from the Instituto Nacional de Ciência e Tecnologia de Inovação em Doenças de Populações Negligenciadas (INCT-IDPN). A.C. was supported in part by NIH grants AI052733, AI15207, and HL059842.

We thank the Johns Hopkins University School of Medicine Microscope Facility for the transmission electron microscopy (TEM) of EVs. Parts of this work were performed in the Environmental Molecular Science Laboratory, a U.S. Department of Energy (DOE) national scientific user facility at PNNL in Richland, WA.

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
