## [Reviewer comments · mSystems]

Comparative molecular and immunoregulatory analysis of extracellular vesicles from *Candida albicans* and *Candida auris*

Daniel Zamith-Miranda, Heino Heyman, Sneha Couvillion, Radames Cordero, Marcio Rodrigues, Leonardo Nimrichter, Arturo Casadevall, Rafaela AmatuZZi, Lysangela Alves, Ernesto Nakayasu, and Joshua Nosanchuk

Corresponding Author(s): Joshua Nosanchuk, Albert Einstein College of Medicine

Review Timeline:

Submission Date:

June 25, 2021

Accepted:

July 30, 2021

Editor: Frank Schmidt

Reviewer(s): The reviewers have opted to remain anonymous.

Transaction Report:

DOI: <https://doi.org/10.1128/mSystems.00822-21>

July 30, 2021

Dr. Joshua D Nosanchuk
Albert Einstein College of Medicine
Microbiology and Immunology
Bronx

Re: mSystems00822-21 (Comparative molecular and immunoregulatory analysis of extracellular vesicles from *Candida albicans* and *Candida auris*)

Dear Dr. Joshua D Nosanchuk:

Your manuscript has been accepted, and I am forwarding it to the ASM Journals Department for publication. For your reference, ASM Journals' address is given below. Before it can be scheduled for publication, your manuscript will be checked by the mSystems senior production editor, Ellie Ghatineh, to make sure that all elements meet the technical requirements for publication. She will contact you if anything needs to be revised before copyediting and production can begin. Otherwise, you will be notified when your proofs are ready to be viewed.

As an open-access publication, mSystems receives no financial support from paid subscriptions and depends on authors' prompt payment of publication fees as soon as their articles are accepted. =

Publication Fees:

We recognize that the video files can become quite large, and so to avoid quality loss ASM suggests sending the video file via <https://www.wetransfer.com/>. When you have a final version of the video and the still ready to share, please send it to Ellie Ghatineh at eghatineh@asmusa.org.

Sincerely,

Frank Schmidt
Editor, mSystems

Journals Department
Table S4: Accept
Table S1: Accept
Fig. S1: Accept
Table S3: Accept
Table S2: Accept
Fig. S2: Accept